# Carbone Monoxide (CO) Detection Device Based on the Nickel Antimonate Oxide and a DC Electronic Circuit

**José Trinidad Guillen Bonilla** [1,2,*] **, Héctor Guillén Bonilla** [3],
**Verónica María Rodríguez Betancourtt** [4]**, Antonio Casillas Zamora** [3],
**Jorge Alberto Ramírez Ortega** [3]**, Lorenzo Gildo Ortiz** [5]**, María Eugenia Sánchez Morales** [6],
**Oscar Blanco Alonso** [5] **and Alex Guillén Bonilla** [7,*]

[1] Departamento de Electrónica, Centro Universitario de Ciencias Exactas e Ingenierías (C.U.C.E.I.), Universidad de Guadalajara, Blvd. M. García Barragán 1421, Guadalajara 44410, Jalisco, Mexico

[2] Departamento de Matemáticas, Centro Universitario de Ciencias Exactas e Ingenierías (C.U.C.E.I.), Universidad de Guadalajara, Blvd. M. García Barragán 1421, Guadalajara 44410, Jalisco, Mexico

[3] Departamento de Ingeniería de Proyectos, Centro Universitario de Ciencias Exactas e Ingenierías (C.U.C.E.I.), Universidad de Guadalajara, Blvd. M. García Barragán 1421, Guadalajara 44410, Jalisco, Mexico

[4] Departamento de Química, Centro Universitario de Ciencias Exactas e Ingenierías (C.U.C.E.I.), Universidad de Guadalajara, Blvd. M. García Barragán 1421, Guadalajara 44410, Jalisco, Mexico

[5] Departamento de Física, Centro Universitario de Ciencias Exactas e Ingenierías (C.U.C.E.I.), Universidad de Guadalajara, Blvd. M. García Barragán 1421, Guadalajara 44410, Jalisco, Mexico

[6] Departamento de Ciencias Tecnológicas, Centro Universitario de la Ciénega (CUCienéga), Universidad de Guadalajara, Av. Universidad No. 1115, LindaVista, Ocotlán C.P. 47810, Jalisco, Mexico

[7] Departamento de Ciencias Computacionales e Ingenierías, Centro Universitario de los Valles (CUValles), Universidad de Guadalajara, Carretera Guadalajara-Ameca Km. 45.5, Ameca 46600, Jalisco, Mexico

* Correspondence: trinidad.guillen@academicos.udg.mx (J.T.G.B.); alexguillenbonilla@gmail.com (A.G.B.); Tel.: +52-(375)-7580-500 (ext. 47417) (A.G.B.)

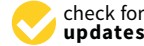

**Featured Application: Our CO detection device finds practical safety applications where there is possible leakage of carbon monoxide by combustion and it is desirable to detect it. For example: cracked heat exchangers, locked chimneys, boiler safety systems, inadequate installation of boilers, heating devices without exhaust gas ducts and inverted fireplace effect.**

**Abstract:** Carbon monoxide (CO) is very toxic to health. CO gas can cause intoxication and even death when the concentration is high or there are long exposure times. To detect atmospheres with CO gas concentration detectors are placed. In this work, a novel CO detection device was proposed and applied for CO detection. For its implementation, four stages were developed: Synthesis of nickel antimonite ($NiSb_2O_6$) oxide powders, physical characterization of $NiSb_2O_6$ powders, Pellet fabrication and sensitivity test in CO atmospheres and electronic circuit implementation where signal adaptation and signal amplification were considered. Experimentally, a chemical sensor was built and characterized, its signal adaptation circuit was implemented and also it was proved using CO concentrations from 1 to 300 ppm with the operating temperatures of 100, 200, and 300 °C. Its optimal operation was at 300 °C. From the experimental results, the CO detection device had excellent functionality because the chemical sensor based on the nickel antimonite oxide had high sensitivity and good electrical response, whereas the DC electronic circuit had good performance.

**Keywords:** a novel CO detection device; CO detection; nickel antimonite oxide; chemical sensor; electronic circuit implementation; high sensitivity

## 1. Introduction

Air pollution produced by the emission of gases such as CO, $CO_2$, $CH_4$, $C_3H_8$, $SO_2$, and $H_2S$, among others [1–3], has caused a substantial decrease in the quality of life of humans and other species. This emission of toxic gases into the atmosphere of large cities by industries and internal combustion engines is considered to be the main cause of diseases due to respiratory and cardiovascular nature, not to mention the greenhouse effect and the ecological and environmental imbalance they also provoke [4–9]. To mitigate the damage, it is necessary to have better control of the concentrations of harmful gases. For this purpose, different binary semiconductor oxides, such as $SnO_2$, $Mn_2O_3$, $GeO_2$, $MoO_3$, and $Nb_2O_5$, among others, have been investigated in the past [10–12]. It has been recently suggested that ternary semiconductor oxides are strong candidates to be applied as gas sensors [13–16]. Among these materials, we can find the perovskite (e.g., $LaFeO_3$) [17] and the spinel (e.g., $CuFe_2O_4$) type structures [18]. On the other hand, other studies report that semiconductor oxides with trirutile-type structure possess interesting detection properties since they are chemically stable and can operate at different temperature ranges (mainly between 200 and 350 °C) [19]. The trirutile-type oxides most commonly employed for this application are $CuSb_2O_6$ [20], $NiSb_2O_6$ [21], $MgSb_2O_6$ [22], $ZnSb_2O_6$ [23], and more recently $CoSb_2O_6$ [24].

It has been established that the high efficiency and good response of semiconductor materials for detecting toxic gases are mainly due to their morphology and porosity, and the nanometric size of the particles [19,22]. To obtain this type of microstructural features, different synthesis routes have been used in the past, like the solid-state reaction, hydrothermal, coprecipitation, and sol-gel processes [25–27]. On the other hand, the colloidal synthesis method has been increasingly employed because through this process it is possible to obtain porous surfaces, diverse morphologies (nanoparticles, nanowires, nanorods, and nanobelts), and particle sizes less than 100 nm [22–24].

The $NiSb_2O_6$ oxide was applied for the Liquefied Petroleum Gas (LPG) and Carbon dioxide ($CO_2$) detection. The oxide was synthetized using a via sol-gel spin coating method. In the experimental work, the electrical resistance changed when the $NiSb_2O_6$ oxide was exposed to LPG and $CO_2$. The measurement concentrations were into the interval of 1000 until 5000 ppm [21]. In reference [28], the authors proposed a chemical detector based on $NiSb_2O_6$ oxide. The $NiSb_2O_6$ powders were synthetized using the microwave-assisted colloidal method. In the experiments, propane ($C_3H_8$) and Carbon monoxide (CO) were detected, the concentrations were between 0 and 300 ppm, the operation temperatures were from 100 to 300 °C and the maximum sensitivity was approximately 2.14 (to 300 °C). On the other hand, using the microwave-assisted wet chemical method, the chemical sensor based on the $NiSb_2O_6$ oxide was better. The sensitivity was increased from 2.1 to 14.1 (to 300 °C) [29]. Again, in the experimental work, $C_3H_8$ and CO were used and the concentrations were between 0 and 300 ppm. However, to our knowledge, the signal adaptation was not proposed for the chemical sensor based on the $NiSb_2O_6$ oxide. In this work, the nickel antimonite ($NiSb_2O_6$) oxide was synthetized, characterized and applied as a chemical sensor for the carbon monoxide detection. Experimentally, its signal adaptation was implemented using a DC electronic circuit which was based on a Wheatstone bridge and operational amplifiers. The prototype device can detect CO concentration into the interval of 1 until 50 ppm and its optimal operating temperature was 300 °C. Our CO detection device has easy implementation, low cost, high sensitivity, good functionality, and also many practical safety applications.

## 2. CO Device Implementation

To build the CO detection device four stages were required: Synthesis of $NiSb_2O_6$ oxide powders, physical characterization of $NiSb_2O_6$ powders, Pellet fabrication and sensitivity test in CO atmospheres, Signal adaptation and signal amplification where the DC electronic circuit was supplied with a DC supply voltage. Figure 1 shows the four stages. Each stage is described in the next sections.

Materials

**Figure 1.** Stages required for the CO detection device.

### 2.1. Synthesis of NiSb$_2$O$_6$ Oxide Powders

The production of the trirutile oxide powders was done through a microwave-assisted wet-chemistry process [28–30] using 1.4560 g of Ni(NO$_3$)$_2$·6H$_2$O (Aldrich, 99%), 2.2802 g of SbCl$_3$ (Sigma-Aldrich ≥ 99%), and 0.5 mL of ethylenediamine (Sigma-Aldrich ≥ 99%). The reagents were diluted in 5 mL of absolute ethyl alcohol (Jalmek) and then left stirring for 20 min at room temperature. The solutions of nickel nitrate and antimony chloride were added dropwise to the ethylenediamine solution. The resulting mixture produced a green precipitate, which was kept under stirring at room temperature for 24 h at a speed of 300 rpm. Evaporation of the solvent was done by applying seventeen 90-s exposures to a low power (140 W) microwave radiation using a domestic microwave oven (General Electric, model JES796WK). The energy absorbed by the solution was estimated at ~214.2 kJ. Subsequently, using a programmable Novatech muffle, two thermal treatments at a rate of 100 °C/h in the air were applied to the precursor material: A drying at 200 °C for 8 h and calcination at 800 °C for 5 h.

### 2.2. Physical Characterization of NiSb$_2$O$_6$ Powders

The characterization of the crystalline phase of the trirutile oxide NiSb$_2$O$_6$ was carried out using powder X-ray diffraction (XRD). The analysis was done using Panalytical Empyrean equipment with CuK$\alpha$ radiation and a wavelength of 1.546 Å. The diffractograms were obtained at a 2θ scan from 10° to 60°, applying 0.026°-steps at a rate of 30 s per step. The porosity, morphology, and particle size of the material dried at 800 °C were analyzed using a scanning electron-microscope (SEM, JEOL JSM-6390LV) in modality of high vacuum and secondary electron emission.

### 2.3. Pellet Fabrication and Sensitivity Test in CO Atmospheres

The sensitivity tests consisted of measuring the changes in the NiSb$_2$O$_6$ pellets' electrical resistance at different operating temperatures (100, 200, and 300 °C) and different concentrations of CO (0, 5, 50, 100, 200, and 300 ppm). The pellets of the material were elaborated using 0.3 g of the powder dried at 800 °C and compacting it by means of a hydraulic press (Simplex Ital Equip-25 Tons) applying 10 tons for 60 s. The dimensions of the pellets were of 0.5 mm in thickness and 12 mm in diameter. Prior to the gas detection measurements, two ohmic contacts of colloidal silver paint (Alfa Aesar, 99%) were placed on the surface of the pellets. Afterward, the NiSb$_2$O$_6$ pellets were placed inside a measuring chamber (Chambers) with a vacuum capacity of $10^{-3}$ torr. The partial pressure of the gases inside the

chamber was controlled using a TM20 Leybold detector. The electrical resistance measurements were made using a Keithley 2001 multimeter. Finally, the response (sensitivity) was estimated according to the formula [20,22,24]: S = $(G_G−G_O)/G_O$, where, $G_G$ y $G_O$ are the pellets' conductances (1/electric resistance) in the test gas (CO) and air, respectively.

### 2.4. Electronic Circuit

For the signal adaptation of our chemical sensor, an economic electronic circuit was proposed. The electronic circuit consists of a Wheatstone bridge, a differential amplifier and an operational amplifier (Op-amp) comparator have been applied. The electronic circuit has been supplied by Vcc1 and Vcc sources. The Wheatstone bridge consists of four resistors: $R_1 = R_2$ are precision resistors, $R_s$ is the resistance of the chemical sensor and $R_x$ is a variable resistance for calibration. The differential amplifier consists of an operational amplifier and four precision resistors, $R_3$, $R_4$, $R_5$, and $R_6$. Finally, the Op-am comparator will be an operational amplifier. Basically, the signal adaptation circuit requires two stages: calibration and detection.

### 2.4.1. Signal Adaptation Circuit (Wheatstone Bridge Calibration)

In the calibration stage, the chemical sensor is installed in atmospheres with no presence of CO, the sensor terminals are connected to the Wheatstone bridge according to Figure 2 and the Wheatstone bridge is calibrated by varying $R_x$ until the following condition is met:

$$V_{AB} = V_A − V_B = 0 \tag{1}$$

and then the next condition $V_A = V_B$ is true.

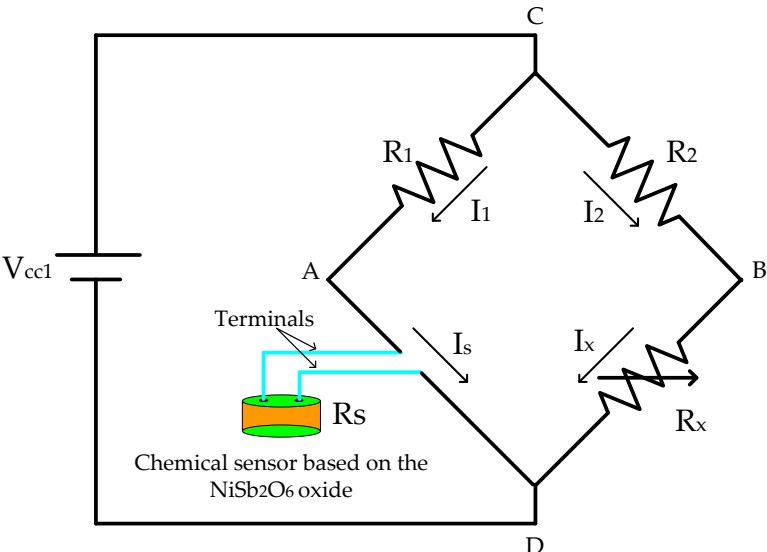

**Figure 2.** The conditions required for the calibration of a Wheatstone bridge.

Based on the Equation (1) and Figure 2, we will have

$$V_A = V_B \rightarrow \frac{V_{CA}}{V_{AD}} = \frac{V_{CB}}{V_{BD}} \tag{2}$$

Observing Figure 2, $V_{CA}$, $V_{AD}$, $V_{CB}$, and $V_{BD}$ are defined as

$$\frac{V_{CA}}{V_{AD}} = \frac{I_1 R_1}{I_s R_s} \ and \ \frac{V_{CB}}{V_{BD}} = \frac{I_2 R_2}{I_x R_x} \tag{3}$$

Since the Wheatstone bridge is calibrated, then $I_1 = I_s$ and $I_2 = I_x$ are true and as a consequence, Equation (2) can express through

$$V_A = V_B \rightarrow \frac{R_1}{R_s} = \frac{R_2}{R_x} \tag{4}$$

From Equation (4), the resistance $R_x$ can be determined by

$$R_x = R_s \frac{R_2}{R_1} \tag{5}$$

When Equation (5) is satisfied, then the Whetstone bridge is completely calibrated.

### 2.4.2. Detection Circuit (Signal Adaptation + Signal Amplification)

In the detection stage, the following five steps are carried out: (a) The sensor terminals are connected to the Wheatstone bridge according to Figure 3, (b) The chemical sensor is exposed to the carbon monoxide atmosphere placing the electronic circuit in a secure area, (c) When the chemical sensor detects the presence of CO, the Wheatstone bridge has an imbalance and the following conditions are met:

$$V_A \neq V_B \; and \; V_A < V_B \tag{6}$$

(d) Both voltages $V_A$ and $V_B$ have been compared applying a differential amplifier based on the operational amplifier, see Figure 3.

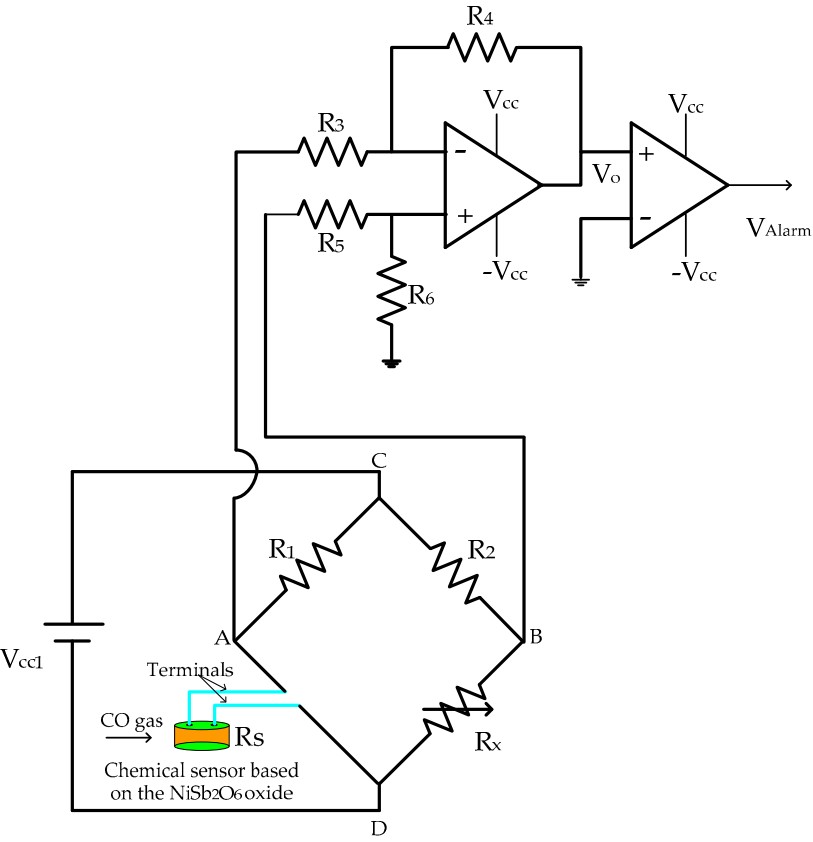

**Figure 3.** Electronic circuit proposed for the CO detection.

From Figure 3, the differential amplifier satisfies

$$V_o = V_B \left( \frac{R_6}{R_6 + R_5} \right) \left( 1 + \frac{R_4}{R_3} \right) - V_A \frac{R_4}{R_3} \tag{7}$$

If $R_3 = R_5$ and $R_4 = R_6$, Equation (7) takes the form

$$V_o = A(V_B - V_A) \qquad (8)$$

where the amplification, $A = \frac{R_4}{R_3} = \frac{R_6}{R_5}$ is a factor and $V_o$ is an output voltage of our differential amplifier. Immediately, we impose the conditions $R_3 = R_4$ and $R_5 = R_6$. Then, the output voltage is the difference

$$V_o = V_B - V_A \qquad (9)$$

(e) The voltage $V_o$ is applied as an input signal to the comparator circuit verifying that:

$$V_{Alarm} = A_{ol}V_o \qquad (10)$$

where $V_{Alarm}$ is the alarm signal produced by the detection device and $A_{ol}$ is the gain of the operational amplifier in an open loop. Thus, the signal is amplified by the operational amplifier. According to Equation (10), the device will generate an alarm signal when the chemical sensor is exposed to carbon monoxide, CO.

## 3. Experimental Results

### 3.1. XRD Analysis

Figure 4 shows the diffractogram of the material obtained from the chemical reaction between the nickel nitrate and the antimony chloride, plus ethylenediamine, and calcined at 800 °C. In the XRD pattern, the characteristic peaks of $NiSb_2O_6$ located at points $2\theta = 19.23°, 21.41°, 27.15°, 33.49°, 34.99°, 38.77°, 40.16°, 43.63°, 44.71°, 53.21°$, and $56.20°$ can be clearly observed.

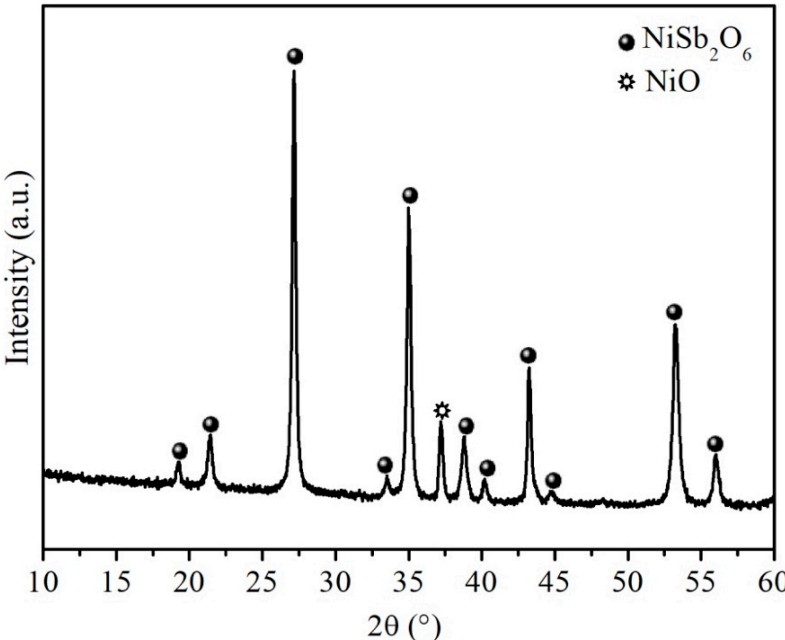

**Figure 4.** Diffractogram of the oxide $NiSb_2O_6$ obtained by calcination at 800 °C in air.

The main crystalline phase of the oxide was identified by means of the PDF file No. 86-0110. According to this, the $NiSb_2O_6$ crystallized into a tetragonal structure with cell parameters a = b = 4.641 Å and c = 9.219 Å, with spatial group $P4_2/mnm$ (136). It has been previously reported that the peaks of the oxide's main phase (see Figure 4) belong to the trirutile-type family of structures or the trirutile-type antimonates [21,28]. This characteristic is obtained by modifying the axis' c-direction of

the rutile cell [31]. On the other hand, a small portion of inorganic material outside the main phase can be seen in the diffractogram. This reflection was located at point 2θ = 37.1° and corresponds to a secondary phase of NiO according to the PDF file No. 65-2901.

The results shown in Figure 4 are consistent with literature reports [32,33] where different synthesis methods of the NiSb$_2$O$_6$'s crystalline phase were employed, like the solid-state reaction [31] and the colloidal method [34]. In this work, an economical and simple to implement alternative, a wet-chemistry process, was used successfully. It was possible to obtain with its particle sizes less than 100 nm.

### 3.2. SEM Analysis

Figure 5 shows three typical photomicrographs of the calcined NiSb$_2$O$_6$'s surface at different magnifications: 500×, 7000×, and 15000×.

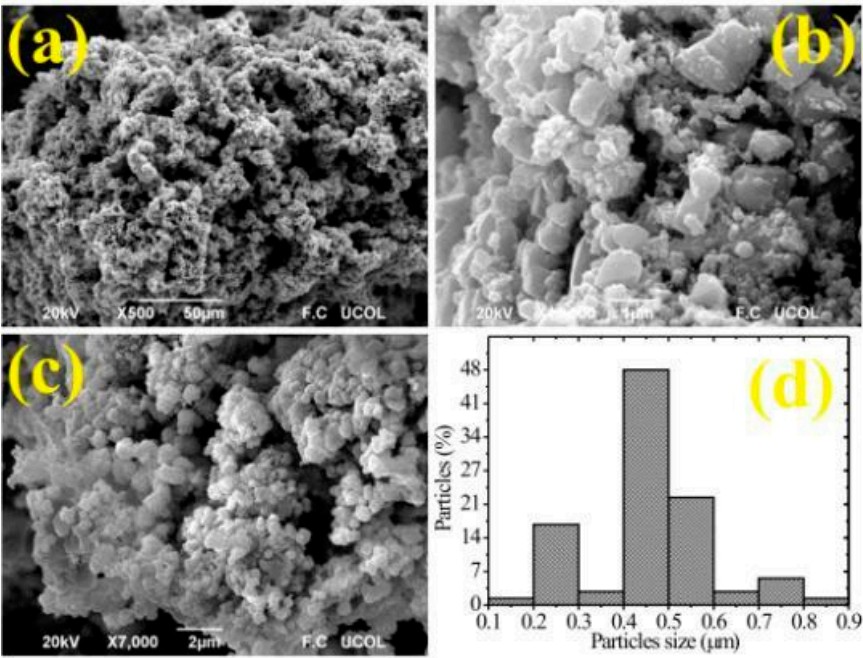

**Figure 5.** SEM images of the calcined (at 800 °C) NiSb$_2$O$_6$'s microstructure at magnifications: (**a**) 500×, (**b**) 7000×, (**c**) 15000×, and (**d**) Particle size distribution.

According to the results shown in Figure 5a,b, a large agglomeration of differently sized particles can be identified. The morphology of the particles is irregular, showing a rough surface generated by the elimination of organic matter when the material is calcined at 800 °C (see Figure 5c). Growth and nucleation of this type of morphology can be attributed to the residence time of the material inside the muffle and to the effect of the ethylenediamine used for the preparation of the NiSb$_2$O$_6$. We have reported in previous works that, by using ethylenediamine during the synthesis process, it was possible to obtain different types of morphologies, like microrods, octahedrons, hexagonal, microplates, and mesoporous-nanoparticles suitable for being used as gas sensors [19,20,22–24,28,29].

The size of the particles was estimated in the range of 0.1–0.9 μm, with an average of ~0.457 μm and a standard deviation of ±0.125 μm. These results are shown in the form of a histogram of the particle size distribution in Figure 5d. The porosity on the material's surface is largely due to the release of gases, such as water vapor, NO$_x$, and CO$_2$ during the heat treatment applied to the material [30].

In general, the ethylenediamine's effect on the production of different morphology types has been thoroughly discussed in previous works [32,33]. The ethylenediamine is incorporated into the inorganic framework and then lost by the heat treatment, thus emerging different types of microstructures [32,33], as occurred in this work. LaMer and Dinegar [35] proposed a mechanism that involves the formation of nanoparticles with different types of morphologies (mainly colloidal dispersions). These authors

refer that the growth and nucleation of inorganic monodisperse colloidal particles are based on three fundamental stages: The first one states that the concentration of the reactants in the colloidal dispersions increases gradually, the second one establishes that the concentration of the reactants reaches a limit of supersaturation and thus, nucleation occurs rapidly forming the nuclei of the crystals, finally, the third one refers that the growth of the particles occurs, originating their morphology. Agreeing with these authors, the microstructure depicted in Figure 5 can be attributed to the production of stable nuclei formed during the reaction of the reactants when the ethylenediamine is added to the process, giving rise to a colloidal dispersion [19] and, as a consequence, the morphology and the particle sizes shown in this work.

### 3.3. Sensing Properties Analysis

Figure 6 shows the response of the $NiSb_2O_6$ pellets as a function of the CO concentration at the operating temperatures 100, 200, and 300 °C. It can be clearly verified in the graph that the pellets are sensitive to the concentrations of the gas. The high response of the material is closely related to the chemical adsorption of the gas on the pellets' surface, as well as to the increase of the test gases' concentration and of the temperature. This is corroborated in Figure 6 since as the concentration of the gas and the operating temperature increase, the efficiency of the pellets to detect the CO concentrations improves substantially. The results obtained in the gas are attributed to the fact that the increase in the operating temperature favors the reaction kinetics between the surface of the pellets and the test gas [20], contributing to a higher oxygen desorption during the adsorption of the test gas [20,22,24,30], which leads to changes in the material's electrical resistance and, consequently, on the high response obtained during the detection tests. The adsorption of different types of oxygen species at high temperatures has been reported in the literature [36]. It can be seen in Figure 6 that the maximum sensitivity magnitudes, ~0.05 and ~0.35, were respectively obtained at 200 and 300 °C for a carbon monoxide concentration of 300 ppm.

The response of the $NiSb_2O_6$ pellets is associated to the mechanism of oxygen desorption at high temperatures [36], which implies that at a temperature lower than 150 °C, the thermal energy is not enough to provoke the oxygen desorption reactions so that no electrical response occurs regardless of the gas concentration. In general, the most abundant oxygen species at temperatures below 150 °C are the $O_2^-$ ions. Above 150 °C (in our case 300 °C) the formation of oxygen species such as the $O^-$ and $O^{2-}$ ions takes place, which are more reactive [22,24,28,30,36]. Furthermore, the increase of the temperature causes a rise in the solid-gas interactions [19,22,30], leading to a good material's response in CO atmospheres.

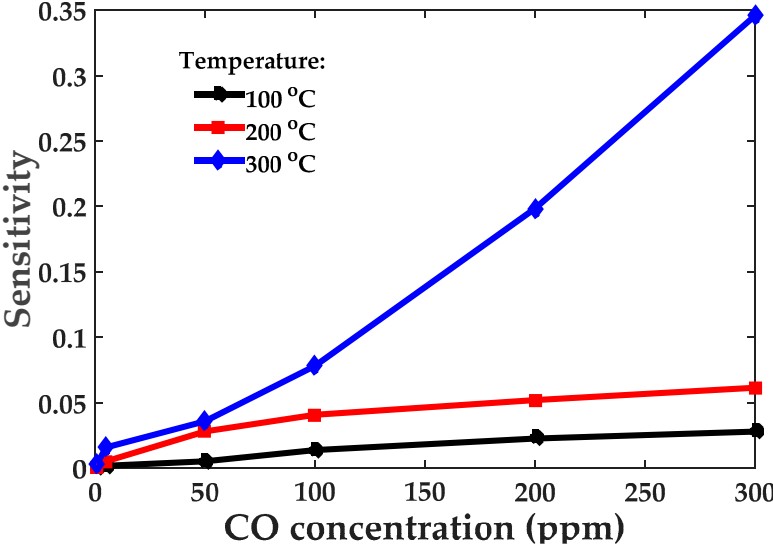

**Figure 6.** Response of $NiSb_2O_6$ pellets as a function of (a) CO and concentrations.

In order to know the variations of the electrical response (resistivity) in the presence of different carbon monoxide (CO) concentrations, the changes in the resistivity were estimated as a function of the gas concentration and the operating temperature. The calculation was made with the formula [20,37]: $\rho = RA/t$, where R is the electrical resistance in the test gases, A is the area of the cross-section, and t is the thickness of the pellets (=0.5 mm, diameter = 12 mm). The results are shown in Figure 7.

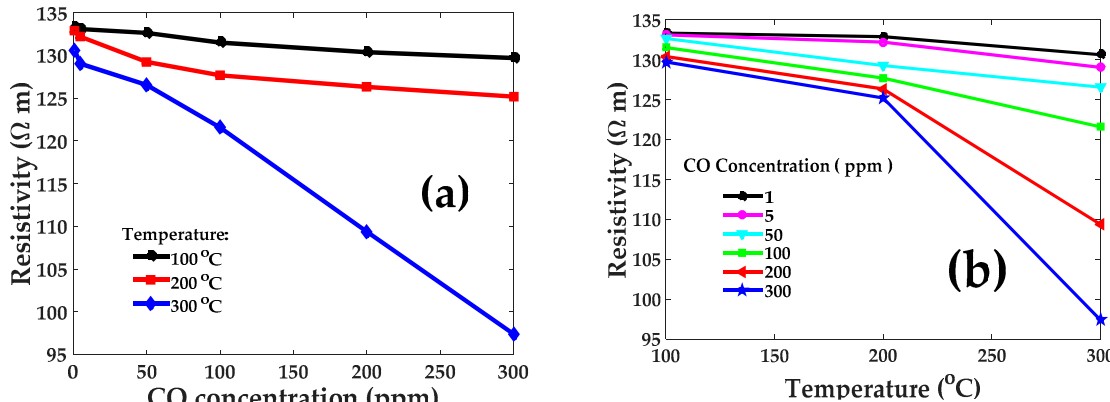

**Figure 7.** Electrical response (resistivity) of $NiSb_2O_6$ pellets as a function of (**a**) CO concentration, (**b**) CO operating temperature.

According to the results, the electrical resistivity of the material in CO atmospheres decays as the concentration and the operating temperature increase. This means that the electrical conductivity of the material increases proportionally as the resistivity decreases (see Figure 7a,b). This behavior is mainly attributed to the fact that, due to the increase of temperature, the charge carriers show more mobility on the surface of the pellets, leading to an increase in the conductivity [20,24]. In addition, it is evident that increasing the concentration of the test gases and the operating temperature, it favors the decrease in resistivity. This phenomenon occurs because the kinetic activity of the gas molecules on the pellets' surface is increased by the rise in operating temperature [24] (in our case at 200 and 300 °C). It has been reported in the literature that the temperature and the geometric shape of the sensor (i.e., pellets or films) plays a quite important role in the chemical interaction between the test gases and the chemisorbed oxygen on the material's surface [20,24,38,39], contributing to producing changes in the electrical resistivity, as shown in Figure 7a,b. This trend is typical of a semiconductor material like the one studied in this work [20,24,37,38,40].

Performing the test on a particle at 300 °C, the resistivity of the material was high, with an almost linear trend regardless of the CO concentrations. However, by increasing the operating temperature to 200 °C, the conductivity increased, and the resistivity decreased. In the case of the oxide pellets, they showed inflection points at 200 °C for both the CO atmospheres (see Figure 7b). The resistivity inflection points shown by the pellets at this temperature are clear indications of the transition that the material undergoes due to the increase in the operating temperature of the sensor. On the other hand, at 300 °C, the resistivity in the atmospheres decreased significantly, where the minimum value of resistivity in carbon monoxide was of 94.0 Ωm at 300 ppm. Considering the results shown in Figure 7a,b, the high response is attributed mainly to the nanometric size obtained during the synthesis process of the oxide. Recent reports have shown that by reducing the particle size of semiconductors, the area-to-volume ratio considerably increases thus favoring high gas adsorption on the surface of the material [30,40,41], leading to a high response in different types of gases.

### 3.4. Resistance vs. CO Concentration

From Figure 7a, the electrical resistance has been calculated for our chemical sensor through Equation $R = \rho t/A$, and then, the behavior of resistance vs. CO concentration can be estimated and graphed as Figure 8 illustrates.

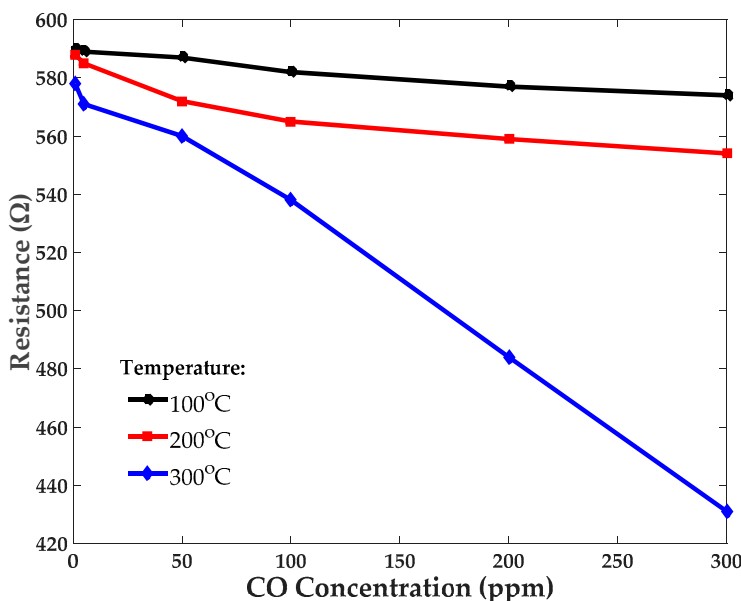

**Figure 8.** Response of NiSb$_2$O$_6$ pellets: Resistance vs. CO concentrations.

Observing Figure 8, when the chemical sensor and CO gas make a reaction, the resistance decays for three temperatures, 100, 200 and 300 °C. The sensor has the highest resistance if both temperatures are 100 and 200 °C. However, when the temperature is 300 °C, sensor's resistance is lower, its decline is major but the sensor has more sensitivity. As a consequence, the operating sensor is better at 300 °C when the sensor´s resistance is into the interval of 578 Ω and 431 Ω.

### 3.5. CO Detection Circuit

Based on Figures 3, 7a and 8, a new CO detection system would be implemented using the NiSb$_2$O$_6$ pellets and a DC electronic circuit. Its characteristics are: its operating concentration is into the interval of 5 and 300 ppm, its operating temperatures are 100 °C, 200 °C, and 300 °C, high sensitivity at 300 °C, economic fabrication and easy implementation. To operate the CO detection system at 300 °C and 50 ppm (or major), we propose the electronic circuit as Figure 9 illustrates.

Observing Figure 9, $R_1$, $R_2$, $R_3$, $R_4$, $R_5$, and $R_6$ have values of 1 KΩ (1000 Ω), the sensor´s resistance $R_s$ was measured to ~560 Ω (Figure 8) and consequently, resistance $R_x$ was calibrated to ~560 Ω (Equation (5)). The Wheatstone bridge is supplied by $V_{cc1} \approx 5$ V. Both operational amplifiers are supplied by $\pm V_{cc} = \pm 12$ V. The differential amplifier has unitary amplification. Finally, the comparator has its gain in the operational amplifier in an open loop, $A_{ol} = 10,000$ (see Equation (10)).

On the other hand, Figure 10 shows the electronic diagram of our DC supply voltage. The voltage source has an error of ±0.2 V, high repeatability, low cost, and good precision. In the power supply, the transformer reduces the alternate current (AC) from 120 V to 18 V, the bridge diode transforms the alternate current (AC) to direct current (DC), the capacitors filter the electrical signal and devices, LM317 T, LM7812, and LM7912 regulate the current voltages, $V_{cc1} \approx 5$ V and $\pm V_{cc} = \pm 12$ V. Notice that, $V_{cc1}$ has a fine-tuning through the resistor $R_{AF}$.

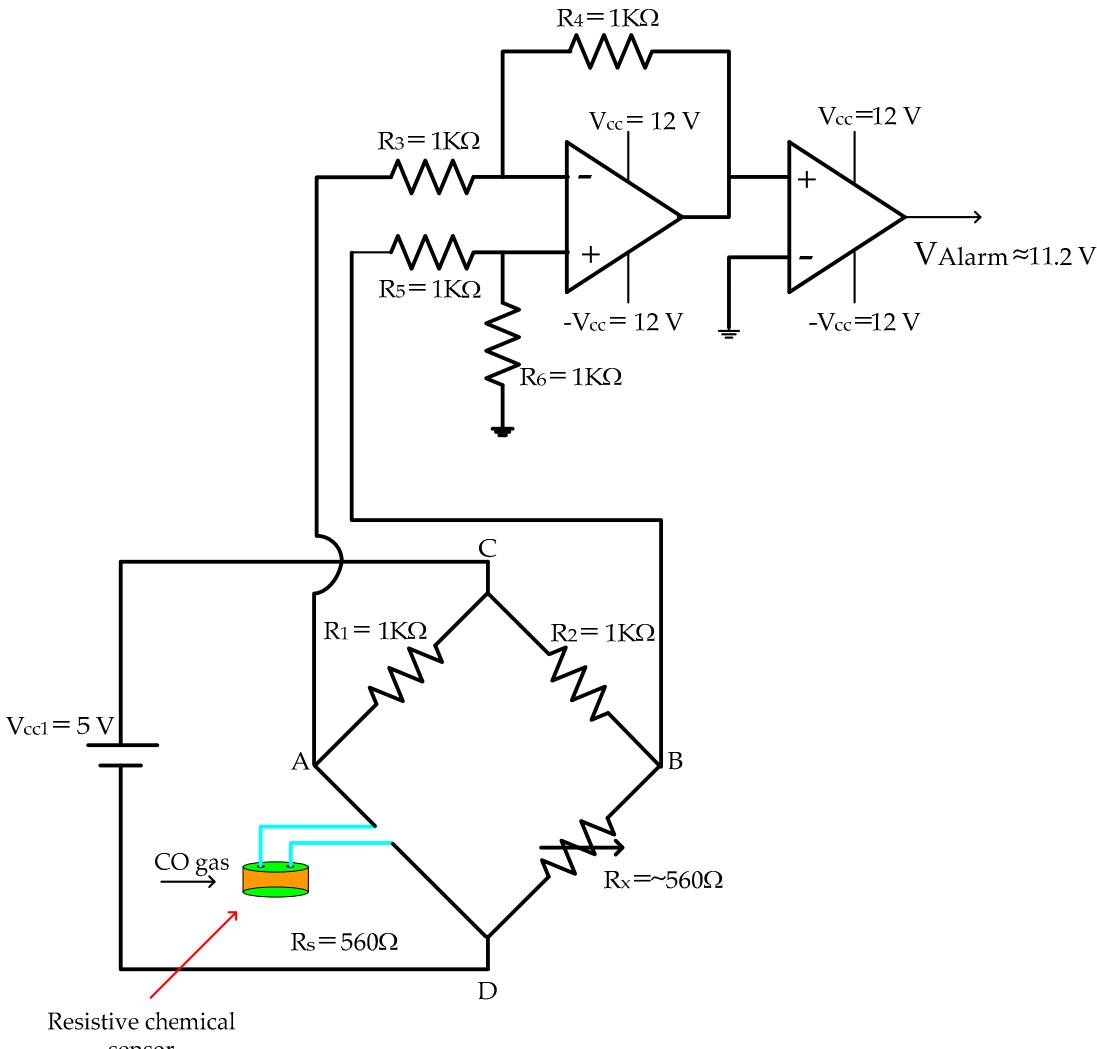

**Figure 9.** A CO detection system proposed at 300 °C and 50 ppm (or higher).

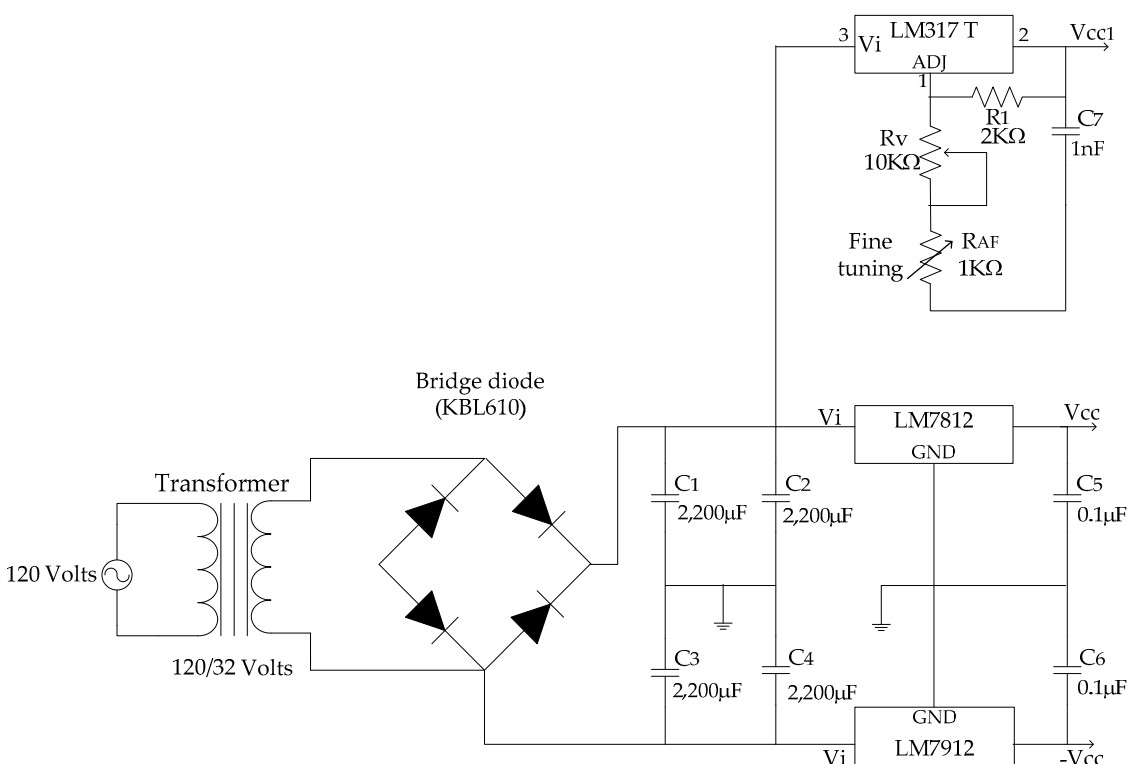

**Figure 10.** Electronic diagram of our DC supply voltage.

### 3.6. CO Detection Device

Figure 11 shows our CO detection device. For its construction, the electronic circuit which was shown in Figure 9 and the DC supply source shown in Figure 10 were considered. The Printed Circuit Board (PBC) was designed using the CAD Altium design program, version 17. Finally, the required materials were: a transformer 120:24 Volts, a diode bridge KBL610 (3A), seven capacitors ($C_1 = C_2, = C_3 = C_4 = 2200$ μF (50 V), $C_5 = C_6 = 0.1$ μF, $C_7 = 1$ nF), seven precision resistors ($R_1 = R_2 = R_3 = R_4 = R_5 = R_6 = I$ KΩ, $R_7 = 2$ KΩ), a potentiometer ($R_v = 10$ KΩ), four rectangular cermet trimmer potentiometers ($R_x = 1$ KΩ, $R_{AF} = 1$ KΩ, $R_{G1} = 10$ KΩ, $R_{G2} = 10$ KΩ), a TL084 (3 V to 32 V, DIP, 14 Pines), a phenolic one side plate (15 cm × 15 cm), three screw connectors, an LM317T (regulator to 5 V), an LM7812 (regulator to 12 V), and an LM7912 (Regulator to −12 V).

In the CO detection device, the sensitivity was increased collocating two resistors, $R_{G1}$ and $R_{G2}$. That is possible because $R_4$ and $R_{G1}$ were connected using a serial array whereas $R_6$ and $R_{G2}$ were also connected to applying a serial array. Therefore, Equation (8) is satisfied if and only if $R_{G1} = R_{G2}$. The output voltage $V_o$ will be

$$V_o = A_m(V_B - V_A) \tag{11}$$

where the modified amplification is given by $A_m = \frac{R_4 + R_{G1}}{R_3} = \frac{R_6 + R_{G2}}{R_5}$ and its value can only be between 1 and 11.

On the other hand, the device operates as Figure 12 illustrates. If CO concentration is equal or greater than 50 ppm, the sensor´s resistance diminishes and as a consequence, the Wheatstone bridge is unbalanced, provoking the signal alarm $V_{Alarm} \approx 11.2$ V (Alarm state "On"). However, if CO concentration is lower than 50ppm, the Equation (6) is not satisfied and then the signal alarm is close to zero, $V_{Alarm} \approx 0$ V (Alarm state "Off"). For our new toxic gas detector device, the thresholding value can be selected through $R_x$: if $R_x > 560$ Ω then the device will detect CO concentrations lower than 50 ppm (or higher) but if $R_x < 560$ Ω then the device will detect CO concentrations greater than 50 ppm.

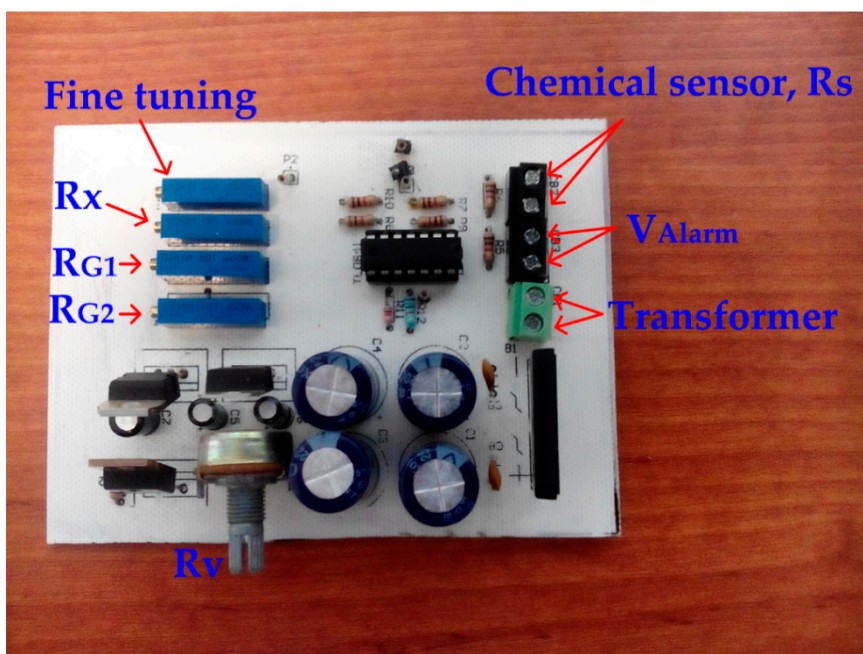

**Figure 11.** CO detection device built and its electronic elements.

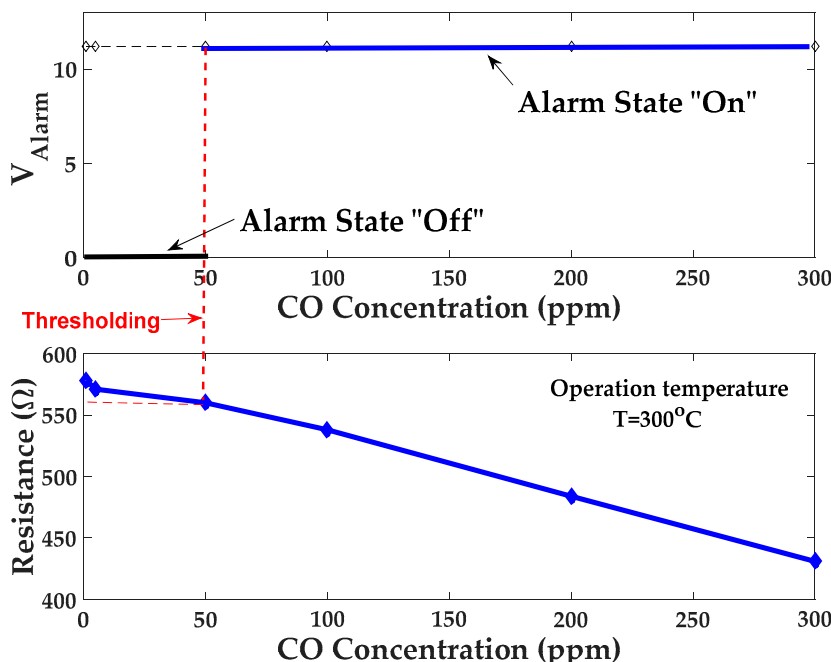

**Figure 12.** Signals measured with our CO detection device (operating temperature to 50 ppm and T = 300 °C).

## 4. Discussion

Based on the experimental results, a new prototype was proposed, developed, and applied for CO gas detection. The prototype device consisted of a chemical sensor and a DC electronic circuit. The chemical sensor was made using the nickel antimonate oxide. The oxide was synthetized by means of an alternative microwave-assisted wet-chemistry process, using nickel nitrate, antimony chloride, ethylenediamine, and ethyl alcohol. The precursor material was dried at 200 °C and later calcined at 800 °C in static air. The crystalline phase of the calcined material was analyzed by X-ray diffraction, finding the cell parameters a = b = 4.641 Å and c = 9.219 Å, and spatial group $P4_2/mnm$

(136). Their sensitivity tests were developed applying static tests (Direct current, DC), an excellent electrical response was obtained. For its signal adaptation a DC electronic circuit was built which was based on a Wheatstone bridge, a DC source, a differential amplifier, and an operational amplifier comparator. That electronic circuit was implemented on a PBC of size 15 cm × 15 cm.

The CO detection device has good characteristics: low cost, high sensitivity, good performance, fast response, selective sensitivity (through trimmer potentiometers), adaptability, CO concentration detected into the interval of 1 to 300 ppm, operating temperatures 100, 200, and 300 °C, optimal operating temperature at 300 °C, its dimension (15 cm × 15 cm), AC supply voltage (120 V) and DC output voltage (11.2 V). However, such characteristics can be improved using better technology for its implementation.

In reference [42], the authors proposed a $CO_2$ detection system based on $CoSb_2O_6$ oxide. For its signal adaptation, the impedance response was analyzed. From the analysis, an electronic circuit was proposed. Its implementation had high complexity because the signal analysis was done on the s-plane, and the electronic circuit was based on the impedance. In this work, another chemical sensor was applied for the CO gas detection but its signal adaptation was made based on the direct current, simplifying the analysis and its implementation. Therefore, comparing both proposals, our proposal reduced complexity, analysis, cost, materials, electronic components and optimized the performance.

The CO detection device finds practical safety applications where CO detection is desirable. Some examples are boiler safety system, locked chimneys, inverted fireplace effect, corroded ventilation pipes and more. Therefore, our future work is in the following direction: CO detector applications, the detector device would be optimized through digital electronics and the nickel antimonate ($NiSb_2O_6$) oxide would be applied for the propane ($C_3H_8$) gas detection.

## 5. Conclusions

We have employed a wet-chemistry synthesis process for producing particles of nanometric size (~23.24 nm on average) for their application in gas detection. This preparation method is economical and very efficient to obtain different morphologies. Furthermore, it is possible to obtain the crystalline phase at relatively low temperatures compared to traditional methods such as the solid-state reaction with this process. The $NiSb_2O_6$ nanoparticles showed high sensitivity in carbon monoxide (CO) atmospheres at different operating temperatures (100, 200, and 300 °C). The optimum performance of the oxide was at concentrations of 300 ppm of CO at 300 °C. The maximum value of the sensitivity was of ~0.35 in CO. Based on these results, a new prototype device was implemented for CO gas detection. Its operating concentration was into the interval of 1 to 300 ppm. Its operating temperatures were 100, 200 and 300 °C. However, its optimal operation was at 300 °C.

The CO detection device was built using analogic electronics: a Wheatstone bridge and electronic circuits based on operational amplifiers. The device had good performance, fast response, selective sensitivity, adaptability, and low cost. Our prototype device finds practical application where it is desirable to detect carbon monoxide leaks.

**Author Contributions:** A.G.B. and J.A.R.O. synthetized the $NiSb_2O_6$ Oxide Powders; H.G.B., V.M.R.B., L.G.O., O.B.A. developed the Physical characterization of $NiSb_2O_6$ Powders; J.T.G.B., A.C.Z., M.E.S.M. developed the signal analysis and proposed the electronic circuit. All authors wrote the paper.

**Funding:** This research received no external funding.

**Acknowledgments:** The authors thank the Mexico's National Council of Science and Technology (*CONACyT*) for the support granted. Antonio Casillas Zamora expresses his gratitude to CONACyT for his scholarship. This investigation was carried out following the line of research "Nanostructured Semiconductor Oxides" of the academic group UDG-CA-895 "Nanostructured Semiconductors" of *CUCEI*, University of Guadalajara. Likewise, we thank Sergio Oliva-León, Miguel-Ángel Luna-Arias, M. L. Olvera-Amador, for their technical assistance.

**Conflicts of Interest:** The authors declare no conflict of interest.

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
