# Peer review of "Carbone Monoxide (CO) Detection Device Based on the Nickel Antimonate Oxide and a DC Electronic Circuit"

_applsci, doi:10.3390/app9183799_

Round 1
Reviewer 1 Report
This manuscript reports the successful synthesis of NiSb2O6 and fabrication of CO sensors. This work is of potential scientific significance, which make it interesting to the readers of Applied Sciences.
Prior to publication, several issues are suggested to be addressed.
The authors emphasize that the sensor response is related to the size of NiSb2O6 grains, so the SEM analysis of the obtained pellets is essential. In order to correctly assess the work of the sensor, you should provide information on response and recovery times (t90% resp and t90% rec, respectively). The proposed devices are to act as a gas detector. Was the aging of the sensor material investigated? Please provide cross-selectivity analysis for the sensor; this will allow readers to assess the applicability of the manufactured device.
Author Response
Subject: Response to reviewers’ comments of paper Ref. No. applsci-586412.
We thank you the reviewers for the valuable comments made to our article, which helped us to improve it. So we are pleased to re-submit for publication the revised version of our paper entitled “Carbone monoxide (CO) detection device based on the nickel antimonite oxide and a DC electronic circuit”, which contains corrections according to all reviewer’s remarks. We hope the manuscript’s updated version meets your expectations.
Reviewer #1:
This manuscript reports the successful synthesis of NiSb2O6 and fabrication of CO sensors. This work is of potential scientific significance, which make it interesting to the readers of Applied Sciences.
Prior to publication, several issues are suggested to be addressed.
The authors emphasize that the sensor response is related to the size of NiSb2O6 grains, so the SEM analysis of the obtained pellets is essential. In order to correctly assess the work of the sensor, you should provide information on response and recovery times (t90% resp and t90% rec, respectively). The proposed devices are to act as a gas detector. Was the aging of the sensor material investigated? Please provide cross-selectivity analysis for the sensor; this will allow readers to assess the applicability of the manufactured device.
Response:
This is a good point. We express our gratitude for your comments. We added lines: 12-123 (page 3) and section 3.2. “SEM analysis”, lines 198 – 248, pages 8 and 9.
We added SEM images to our manuscript. The goal is to show the morphology obtained from the synthetized oxide since it is well documented that gas detection properties are strongly related to the microstructure of a material. Additionally, a histogram was also added for our new manuscript version. The histogram shows the particle size distribution which was measured for the oxide NiSb2O6.
On the other hand; unfortunately, in this work, dynamic tests in CO atmospheres were not considered due to the fact that we don’t currently have a way to perform such experiments. At present, we only have a measurement system to perform response tests versus concentration and working temperature. We are doing everything possible to have additional measurement methods soon.
We express our gratitude for your comments and observations.
Kind regards,
Dr. Alex Guillén-Bonilla
Professor - Research Scientist, Department of Computational Sciences and Engineering, CUValles, University of Guadalajara, Carretera Guadalajara-Ameca Km. 45.5, 46600, Ameca, Jalisco, México
+52 (33) 1378 5900 ext.
alexguillenbonilla@gmail.com.

Reviewer 2 Report
The authors should use spelling check and ask a native English speaker to review and edit this paper. Below are some mistakes I found
Line 33, "corroided" is not an English word.
Do not use upper case letter in the middle eve sentences.
Line 338 COSb2O6 should be corrected to CoSb2O6.
Author Response
Subject: Response to reviewers’ comments of paper Ref. No. applsci-586412.
We thank you the reviewers for the valuable comments made to our article, which helped us to improve it. So we are pleased to re-submit for publication the revised version of our paper entitled “Carbone monoxide (CO) detection device based on the nickel antimonite oxide and a DC electronic circuit”, which contains corrections according to all reviewer’s remarks. We hope the manuscript’s updated version meets your expectations.
Reviewer #2:
The authors should use spelling check and ask a native English speaker to review and edit this paper. Below are some mistakes I found:
Response:
We appreciate your comments and observations.
The manuscript was reviewed by a native English speaker.
Line 33, "corroided" is not an English word.
Response: The word “corroided” was removed.
Do not use upper case letter in the middle eve sentences.
Line 338 COSb2O6 should be corrected to CoSb2O6.
Response: COSb2O6 was change by CoSb2O6; Line 386, page 14.
We express our gratitude for your comments and observations.
Kind regards,
Dr. Alex Guillén-Bonilla
Professor - Research Scientist, Department of Computational Sciences and Engineering, CUValles, University of Guadalajara, Carretera Guadalajara-Ameca Km. 45.5, 46600, Ameca, Jalisco, México
+52 (33) 1378 5900 ext.
alexguillenbonilla@gmail.com.

Round 2
Reviewer 1 Report
The publication may be published in its current form.